# An Exploration of Clinical Characteristics and Treatment Outcomes Associated with Dietetic Intervention in Adolescent Anorexia Nervosa

**DOI:** 10.3390/nu16234117

**Published:** 2024-11-28

**Authors:** Cliona Brennan, Lara Felemban, Ellen McAdams, Kevin Walsh, Julian Baudinet

**Affiliations:** 1Maudsley Centre for Child and Adolescent Eating Disorders, South London and Maudsley NHS Foundation Trust, De Crespigny Park, Denmark Hill, London SE5 8AZ, UK; ellen.mcadams@slam.nhs.uk (E.M.); julian.baudinet@kcl.ac.uk (J.B.); 2Department of Human Nutrition and Dietetics, School of Health Sciences, London Metropolitan University, 166-220 Holloway Road, London N7 8DB, UK; 3King’s College London, 150 Stamford Street, London SE1 9NH, UK; lara.felemban1@nhs.net (L.F.); kevin.walsh@kcl.ac.uk (K.W.); 4Department of Nutrition and Dietetics, North Middlesex University Hospital NHS Trust, Sterling Way, London N18 1QX, UK; 5Institute of Psychiatry, Psychology and Neuroscience, King’s College London, De Crespigny Park, London SE5 8AZ, UK

**Keywords:** anorexia nervosa dietetics, family therapy, eating disorders

## Abstract

*Background:* Although dietitians possess expert knowledge on the interplay between nutrition and health, their specific role in family therapy for anorexia nervosa (FT-AN) remains a topic of debate. Some of the literature indicates insufficient evidence to affirm the impact of dietetic interventions, emphasising variability in outcomes and a need for standardised research. This study aimed to identify the clinical characteristics of adolescents requiring dietetic intervention during FT-AN and to assess differences in clinical outcomes between those receiving dietetic support and those who did not. *Methods:* A retrospective cohort study was conducted at the Maudsley Centre for Child and Adolescent Eating Disorders in London. Patients were selected from electronic records. Inclusion criteria were ICD-10 diagnosis of AN and completion of FT-AN treatment between January 2020 and December 2022. Collected data included weight (kg and %mBMI), eating disorder symptom severity, pre-assessment anxiety (patient and parent), and details of FT-AN sessions (i.e., frequency and amount). The sample was divided into two groups: those who received dietetic input and those who did not. Statistical analyses included Mann–Whitney U tests, χ^2^ tests, independent *t*-tests, and a logistic regression to examine differences at baseline, 4–6 weeks post-assessment, and discharge. *Results:* The study included 92 participants (dietetic group = 33 participants; non-dietetic group = 59 participants). Baseline characteristics were comparable between groups. The logistic regression showed no significant predictors for dietetic input. At 4–6 weeks, those requiring dietetic input exhibited lower %mBMI (83.3% vs. 87.3%, *p* = 0.027) and poorer weight gain (+2.3 kg vs. +3.1 kg, *p* = 0.04). By discharge, weight restoration was similar (92% vs. 93% mBMI, *p* = 0.64), although the dietetic group had more therapeutic treatment sessions (24 vs. 19, *p* = 0.04). *Discussion:* This study found no specific predictors for prioritising dietetic input in young people during FT-AN treatment. While those receiving dietetic support struggled with weight gain early and attended more sessions, both groups achieved similar weight outcomes by the end of treatment. Future research should focus on the timing and content of dietetic interventions, as well as perspectives from patients and caregivers, to better understand their role and impact on cognitive and emotional recovery aspects.

## 1. Introduction

Anorexia nervosa (AN) is an eating disorder (ED) that significantly compromises both physical and mental health, characterised by markedly low body weight, restrictive eating behaviours, an intense fear of weight gain, and excessive preoccupation with body weight [1]. Family therapy for anorexia nervosa (FT-AN) is the current first-line recommended treatment in the UK [2]. Initial phases of treatment for AN focus on medical stabilisation, weight gain if required, and restoring regular, adequate nutrition [3,4,5]. Family-Based Therapy (FBT) for AN is a similar, albeit slightly different family therapy model for adolescent anorexia nervosa more commonly used across the United States (US). Early weight gain, facilitated by swift nutritional rehabilitation, is the cornerstone of the initial stages of recovery from AN, resulting in improved clinical outcomes and shorter treatment duration [6,7]. Studies have identified that greater weight gain during initial treatment (i.e., the first month) predicts full remission after one year [8,9]. Given the significance of prompt weight restoration, ensuring adequate nutritional intake is essential in the treatment of AN. Dietetic interventions are recommended in the treatment of AN by national guidance in the UK as part of the specialist MDT, wherein psychological therapy and medical management are central parts of treatment that are delivered by professionals qualified in psychiatry, psychology, and medicine [5,10]. The collaborative nature of the MDT in FT-AN is fundamental to its efficacy and contextualises the dietitians’ contributions within treatment. Dietitians are skilled in the assessment and management of malnutrition, disordered eating patterns, and nutrition-related deficiencies [11,12]. Dietitians are, therefore, well placed to support nutritional related issues arising from AN and are considered to form a central part of the MDT [11,13].

Although dietitians possess expert knowledge on the interplay between nutrition and health, the specific role of dietitians within FT-AN remains a topic of debate [14]. FT-AN posits that parents are a crucial resource in the recovery process of a young person with AN [4,15,16]. The initial phase of FT-AN focuses on engaging the family and young person (YP), building a collaborative and trusting relationship with the therapist, and providing education on the disorder and the consequences of starvation [3,15]. Subsequent phases involve supporting the family in managing the child’s eating behaviour to ensure improved nutritional intake and steady weight gain, progressively handing control of eating back to the YP, and planning for life post-treatment [4].

FBT emphasises parental involvement in the adolescent’s recovery, focusing on restoring healthy eating behaviours and weight gain [16]. It involves three phases: parents initially controlling the child’s eating, gradual transition of control back to the adolescent, and then addressing normal adolescent development and relapse prevention. In FBT, dietitians typically play a consulting role, providing nutritional expertise to therapists as needed rather than directly engaging with patients [16,17]. In contrast, FT-AN involves a more direct role for dietitians, including providing standard meal plans and participating in therapy sessions when necessary [4]. Both approaches acknowledge the importance of nutritional input but differ in the extent and manner of dietetic involvement. The role of dietetics in both modalities remains under-researched. Similarities and subtle differences between these two treatment models have been recently explored in the literature [18].

Despite this guidance, some studies argue that there is insufficient evidence to conclusively determine the impact of dietetic interventions in adolescent AN, citing variability in study results and a need for more robust, standardised research [19,20]. The limited research on dietetic interventions for ED shows mixed results, with some studies reporting significant improvements in psychopathology and others finding no effect [20]. The inconsistencies in findings and variability in methodologies highlight the need for higher-quality, standardised research on the role of dietitians in ED treatment. This calls for clearer guidelines and more rigorous studies to better understand and optimise this resource in the treatment of AN.

In recent years, the role of dietetics in ED treatment has been explored in adult populations, and the views of multidisciplinary team (MDT) members, clients, and dietitians have been used to develop consensus guidelines in this area [20,21,22]. These advances have enabled the evaluation of dietetic practice in adult ED treatment, the definition of the role that the dietitian plays in the ED MDT, and the development of specific dietetic resources for this client group [23]. However, there is a dearth of research and evidence related to this topic in those under 18 years of age. To date, the only published research on this topic includes one qualitative study exploring clinician views of the dietitian’s role in FT-AN, highlighting the importance of research on this specific age group and therapy modality, as opposed to others.

To bridge this research gap, it has been recommended that dietitians conduct research on the effectiveness of dietetic interventions in AN treatment and identify patient factors that indicate the need for dietetic input [24]. Research in this area is urgently needed to support safe and effective dietetic practice that is evidence-based and delivers good clinical outcomes. This study aimed to investigate the characteristics of adolescents with AN who received dietetic intervention as part of FT-AN treatment and to explore differences in clinical outcomes for these groups, as one step towards evidence based dietetic treatment guidelines in this area.

## 2. Methods

### 2.1. Ethical Approval

Ethical approval for this project was granted by the South London and Maudsley Child and Adolescent Mental Health Services (CAMHS) service evaluation and audit committee (approval number 330 and date 23 August 2023). Given the study’s retrospective design and the vulnerable adolescent population involved, ethical consideration such as ensuring anonymity of participants were ensured.

### 2.2. Study Design

This was a retrospective cohort study conducted at the Maudsley Centre for Child and Adolescent Eating Disorders (MCCAED) of the Maudsley Hospital in London, United Kingdom. This study design was chosen to facilitate a greater sample size than prospective methods would provide. In addition, all participants were recruited post discharge to ensure they had completed the FT-AN treatment and that treatment outcomes could be investigated. As such, a retrospective study design was necessary.

### 2.3. Sample

Inclusion criteria were (a) International Classification Diseases, version 10 (ICD-10) diagnosis of AN, and (b) completion of FT-AN treatment between January 2020 and December 2022. Patients were retrospectively recruited from the electronic notes databases used by the service.

Exclusion criteria were as follows: a diagnosis of an eating disorder that was not AN; the absence of underweight (>95% median BMI); did not complete FT-AN treatment (i.e., were admitted to inpatient services prior to completion of FT-AN).

### 2.4. Data Collection

#### 2.4.1. Demographic and Treatment Characteristics

Demographic and treatment data were manually retrieved from patient records. This included review of electronically recorded notes, clinic letters, and referral information. Data extracted included gender, ethnicity, age at assessment, diagnosis, assessment and discharge date, dates of each FT-AN session, weight, height, and percentage median body mass index (%mBMI), dates of measurement, dietetic entries, and reason for discharge. Patients were identified for dietetic input by the FT-AN therapist based on clinical judgement rather than set criteria %mBMI ≥ 95% at discharge was used as a measure of completed weight restoration in this study. Weights recorded in patient records were measured in the treatment centre by a member of the treatment team. All weights were measured at weekly intervals by clinicians using standard weighing procedures (i.e., weighing participants in light clothing, without shoes, with emptied pockets, and after toileting). The data collector was not blind to the study aims. Measures taken to minimise this bias included inter-rater reliability checks.

#### 2.4.2. Self-Report Measures

Prior to assessment at MCCAED (i.e., up to one week prior to assessment), all young people and parents complete a battery of routine outcome measures. The Eating Disorder Examination Questionnaire (EDE-Q), adolescent version [25], was used to assess the severity of eating disorder symptoms. The EDE-Q has been used widely with young people with available adolescent norms.

The Revised Children’s Anxiety and Depression Scale (RCADS), young person and parent version [26], was used to assess symptoms of anxiety and depression. The RCADS has been used extensively and shown to have good psychometric properties.

### 2.5. Statistical Analysis

IBM SPSS Statistics (Version 29) [27] was used to complete all statistical analysis. The sample was split into two independent groups: (1) participants referred to dietetics who attended at least 1 session, and (2) those who did not receive any dietetic input during their treatment. Groups were compared using independent-sample Mann–Whitney U tests for continuous variables with non-normal distribution, χ^2^ tests for categorical variables, and independent *t*-tests for variables with normal distribution. Where assumptions for the analysis of categorical variables were violated (cell count < 5), Fisher’s exact or likelihood ratio tests were used instead. Effect size was calculated using Cramer’s v, Cohen’s d or r, where r = Z/√N. Bivariate correlations were conducted to examine associations between continuous variables.

Groups were compared at three timepoints (baseline, 4–6 weeks post-assessment, discharge). Baseline and discharge timepoints were used to allow the measurement of change over the course of FT-AN treatment, the 4–6 week timepoint was used given the significance of early weight gain in FT-AN and FBT (first month of treatment). Mean weight change 4–6 weeks after the start of FT-AN treatment was used to determine patients’ initial weight restoration early in treatment. In cases where more than one weight measurement was taken within this 4–6-week timepoint, the first measurement was used. A repeated-measures analysis of variance (ANOVA) was used to determine changes in weight across the three timepoints for the two groups.

## 3. Results

### 3.1. Sample

A total of 125 patients were identified who met the inclusion criteria (Figure 1). Of these, 33 were excluded for the following reasons: (1) did not receive FT-AN (*n* = 4); (2) were referred to tier 4 ED services (*n* = 9); (3) were not diagnosed with AN (*n* = 13); (4) had a %mBMI > 95% at assessment (*n* = 2); (5) did not complete FT-AN by 2022 (*n* = 5).

### 3.2. Referrals to Dietetics

Of the 33 patients referred for dietetic input during treatment, referrals were made 51 times, with 11 (33.3%) patients receiving dietetic input more than once. Of these 51 referrals, the most frequent referral reasons were increasing oral intake—this included patients who required specific dietary guidance on increasing their oral intake above the amount prescribed on generic meal plans(*n* = 12, 23.5%), difficulty gaining weight (*n* = 10, 19.6%)—including patients whose weekly weight restoration was consistently below 0.5kg/week, and eating for weight maintenance or normal/intuitive eating—including patients who no longer needed a strict meal plan for weight gain due to being weight restored or in phase 3–4 of FT-AN (*n* = 10, 19.6%). Other reasons for dietetic referrals included assessing nutritional needs or intake (*n* = 5, 9.8%); nutrition psychoeducation (*n* = 5, 9.8%); personalising meal plans (*n* = 2, 3.9%); guidance on portions (*n* = 2, 3.9%); eating for increased physical activity/exercise (*n* = 2, 3.9%); and special dietary requirements (*n* = 2, 3.9%).

### 3.3. Group Characteristics

Table 1 displays the characteristics of the 92 children and adolescents included in the analyses. At the time of assessment, the median age was 15 years, 94.6% were female, 52.2% had AN-R, and 16.3% had AN-BP. The patients were mainly of white ethnicity (*n* = 48, 52.2%), followed by Asian British (*n* = 6, 6.5%) and Black British (*n* = 5, 5.4%), respectively.

Of the 92 patients, 33 (35.9%) were referred for dietetic input during FT-AN and 59 (64.1%) were not. These two groups were similar in all demographic characteristics. Anthropometry data were available for all patients at baseline. Patient weight did not differ significantly between dietetic and non-dietetic groups (81% vs. 80.4% %mBMI, respectively, *p* = 0.588). See Table 1 for more details.

### 3.4. Eating Disorder Symptom Severity, Anxiety, and Dietetic Input

At baseline, there were no differences between groups in the severity of eating disorder symptoms (EDE-Q global score) and parent and young person rated anxiety (RCADS-P and RCADS-YP total anxiety score, respectively). Significant moderate associations were observed in both groups between the severity of eating disorder symptoms and anxiety scores (dietetic: r = 0.59, *p* < 0.001; non-dietetic: *r* = 0.55, *p* < 0.001), which did not significantly differ by strength of association (z = 0.26, *p* = 0.80).

### 3.5. Baseline Predictors of Dietetic Input

None of the baseline and pre-assessment measures were found to significantly predict dietetic input during FT-AN (χ^2^ = 5.45, *df* = 4, *p* = 0.24) (see Table 2).

### 3.6. Clinical Outcomes

#### 3.6.1. FT-AN Treatment

Patients in the dietetic group received dietetic input at various timepoints during FT-AN treatment. A total of 13 patients received dietetic input during early treatment (i.e., first 4–6 weeks of FT-AN treatment), whilst the remaining 20 patients received this input at later stages of FT-AN treatment. Although the average length of FT-AN treatment was not significantly different between groups [dietetic 11.55 months (*SD* = 5.59) vs. non-dietetic 10.12 months (*SD* = 4.65), *t* (90) = −1.32, *p* = 0.19], the total number of FT-AN sessions was significantly greater on average in the dietetic group [24.2 sessions (*SD* = 12.4) vs. 18.7 sessions (*SD* = 10.2), *t* (90) = −2.29, *p* = 0.03, *d* = 0.50]. Six patients who had not completed treatment were discharged due to turning 18 and referred to adult ED services to continue AN treatment (dietetic *n* = 4, non-dietetic *n* = 2), (χ^2^ = 2.65, *p* = 0.10).

#### 3.6.2. Weight Outcomes

Table 3 and Figure 2 displays weight outcomes for both groups during early treatment (initial 4–6 weeks) and point of discharge. Anthropometry data were available for 77.2% (*n* = 71) at the early treatment point and for all patients at baseline and discharge (*N* = 92). A repeated-measures ANOVA demonstrated there was a significant main effect for weight, with both groups increasing weight across all three timepoints (F(1, 2) = 111.07, *p* < 0.001). The weight-by-group interaction effect was not significant (F(1, 2) = 26.42, *p* = 0.387).

Post hoc comparisons demonstrated that patient weight in early treatment differed significantly between groups, with a lower mean weight observed in the dietetic group (%mBMI 83.3% vs. 87.3%, *p* = 0.027). Early weight change from baseline to early treatment also differed significantly between groups, with a lower mean weight gain observed in the dietetic group (+2.3 kg vs. +3.1 kg, *p* = 0.044). At discharge, the mean patient weight was not significantly different between dietetic and non-dietetic groups (92.3% vs. 93.1%, *p* > 0.05), with no significant differences observed between groups in overall %mBMI change.

#### 3.6.3. The Timing of Dietetic Input and Early Weight Gain

For those who received dietetic input, Fisher’s exact test was used to determine if there was a significant association between early weight gain (≥2 kg at early treatment point; yes or no) and the timing of dietetic input (before or after early treatment point). The association between the two variables was not significant (two-tailed *p* = 0.702).

## 4. Discussion

This study explored the characteristics of adolescent patients requiring dietetic input during FT-AN and the associated effect on their clinical outcomes. The study resultsindicate that eating disorder symptom severity, weight, and anxiety are not associated with being referred for dietetic input during FT-AN. Baseline characteristics of those who received dietetic support were comparable to those who did not. Given the fact that dietetic referral was based on clinical judgement rather than set criteria, the variability in clinician decision making or differences in initial patient assessments may have impacted this finding.

These findings highlight two significant differences between the group that did receive dietetic input and the group that did not. Firstly, the group that received dietetic input were less likely to have gained weight early in treatment (i.e., leading to the referral to dietetics, rather than the dietetic input causing poor early weight gain). Secondly, they had significantly more intensive treatment, with an average of 6 additional FT-AN sessions (24 sessions vs. 18 sessions) compared to those not referred for dietetics. Encouragingly, there were no differences in weight outcomes at the end of treatment, suggesting the increased intensity and dietetic input may have been important factors in supporting the group who did not respond early in treatment to reach an equivalent outcome by discharge. This is key as the previous literature has consistently found that inadequate early weight gain is predictive of poorer end-of-treatment outcomes [8,9]. This study’s findings suggest that intensifying treatment and adding dietetic support may address this effectively. One way to understand this finding is that the additional intensity and more multidisciplinary professionals involved in the young person’s treatment helps increase knowledge, confidence, and hope, and contributes to a stronger therapeutic alliance with the multidisciplinary team, a factor identified by young people that helps generate change in FT-AN [28]. The role of specific dietetic interventions in strengthening therapeutic alliance remains under-researched, however we hypothesise that dietetic intervention during FT-AN likely increases trust in the treatment and maximises the delivery of patient centred, individualised care.

Most dietetic referrals were reported by clinicians to be due to challenges in weight restoration and increasing nutritional intake. Previous studies in adults with ED have highlighted that clinicians’ decisions to involve dietitians are typically influenced by a patient’s weight status and treatment progress [29]. Findings from this exploratory study suggest that similar criteria guide clinicians’ decisions with adolescents in FT-AN. The current findings also suggest clinician decision to refer to dietetic input is not necessarily influenced by *early* weight gain, as might have been expected, as the association between early weight gain and the timing of dietetic input was not significant. Rather, clinicians seem to be referring to dietetic support at any point throughout treatment and not always if the patient is struggling with early weight gain. This suggests the inclusion of dietetic involvement in FT-AN is more nuanced than just a lack of early weight gain, and may differ across different treatment stages in FT-AN. Further research is needed to understand this decision-making process more and the impact on outcome.

Indirect dietetic interventions, such as standardised meal plans and professional consultation, are often indicated for young people as part of FT-AN [30]. Meal plans can support adequate weight restoration and containment of the system in the early stages of FT-AN. The delivery of direct dietetic intervention (i.e., direct consultation with parents/carers and young people) depends on individual needs [20,31]. Patient characteristics, previously evidenced to necessitate direct dietetic intervention, include the management of refeeding syndrome [32,33], specialist nutrition support for co-morbidity [29], and tailored advice for special dietary requirements during treatment (such as allergy, intolerance, athletes etc.) [11]. This study’s findings highlight the potential benefits of additional direct dietetic input for young people who are struggling to gain weight or make changes in treatment. Given the importance of prompt early weight gain during initial FT-AN treatment, the inclusion of dietetics in FT-AN could be considered as one possible tool to improve the weight gain trajectory.

Our findings suggest that greater treatment intensity, including the involvement of dietitians, may support better clinical outcomes for patients struggling to restore weight early in treatment. This fits with findings from a recent systematic review that dietetic involvement in ED treatment significantly improved BMI and weight in adults with AN [20]. Heafala et al. [31] also identified dietitians as key contributors on nutritional aspects in AN, emphasising their critical contribution to supporting weight restoration during ED treatment. Developmental differences that exist between adults and young people may influence the impact of dietetic intervention on weight related outcomes in those under 18 years compared with adults (e.g., additional energy needs during puberty may lead to slower weight gain). Additional factors that were not captured by this study, such as readiness to change, motivation, engagement and therapeutic alliance, impact progress in recovery and treatment. Therefore, although a greater intensity of treatment, including dietetic input, may facilitate positive change, a holistic approach must be taken by the treating team to ensure good clinical outcomes [34,35,36].

## 5. Limitations

The results of this study should also be interpreted with caution due to its limitations. Only weight data are reported. While this is an important aspect of recovery, it is not the only factor. Future research is needed to understand the impact of dietetic input on the cognitive and emotional aspects of eating disorder recovery. Additionally, weights and anxiety scores were not available for all patients, which may have affected data analysis by reducing statistical power or the representativeness of the sample. Potential confounding variables that could impact weight restoration outcomes, such as prior treatment history, comorbid mental health conditions, or other support services accessed outside of FT-AN were not captured by this study and may have impacted results. Furthermore, some did not have a weight recorded at discharge, resulting in the use of their last recorded weight, which was sometimes several weeks before discharge date. While this fits clinically with stopping weighing during the final sessions of FT-AN, it is possible that some patients’ weights were different to the weight used in data analysis. This can limit the findings of this study, given that missing weight data at discharge could skew results.

## 6. Conclusions

Based on this study’s findings, there are no patient- or family-specific characteristics at assessment that indicate those who may benefit from dietetic input. No significant predictors for dietetic input were identified in this study, with baseline characteristics of both groups being similar. Nevertheless, those who did receive dietetic input during FT-AN were significantly more likely to have struggled to gain weight early in treatment and attended a greater number of FT-AN sessions. By the end of treatment, however, both groups had equivalent weight outcomes. Dietetic input may support patients struggling with early weight gain, highlighting its value in maintaining equivalent final outcomes. Future research is needed to investigate the timing and content of dietetic interventions, alongside patient and caregiver perspectives, to clarify their role in FT-AN treatment. Family perspective may would valuable insights into the benefits of dietetic input that extend past weight restoration during FT-AN. Further research is needed to explore the impacts of dietetic input on the cognitive and emotional aspects of recovery.

## Figures and Tables

**Figure 1 nutrients-16-04117-f001:**
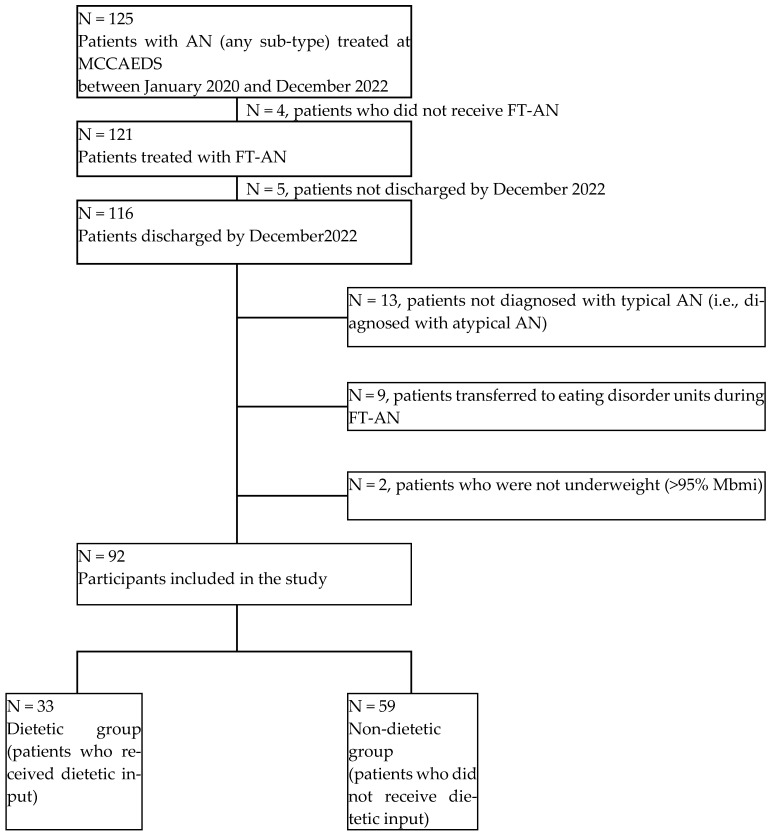
Flowchart of participants included and excluded in study.

**Figure 2 nutrients-16-04117-f002:**
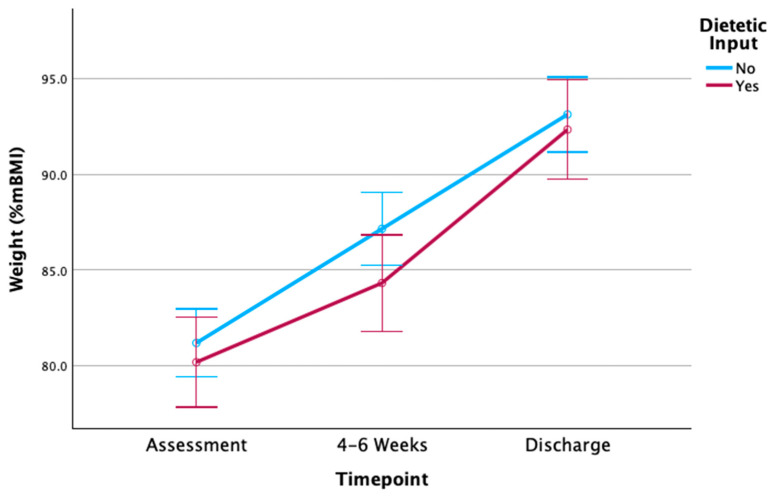
Mean weight (%mBMI) at baseline, early treatment point, and discharge. Note: error bars represent 95% confidence intervals.

**Table 1 nutrients-16-04117-t001:** Baseline characteristics of total sample and comparison of characteristics of groups based on dietetic input during FT-AN treatment.

Characteristic	Total (*n* = 92)	+ Dietetics (*n* = 33)	− Dietetics (*n* = 59)	Test
	Mean (*SD*)	Mean (*SD*)	Mean (*SD*)	
Age (years), Mean (SD)	14.58 (1.77)	14.51 (1.86)	14.61 (1.73)	*z* = 0.50, *p* = 0.96, *r* = −0.01
Missing	0 (0.0%)	0 (0.0%)	0 (0.0%)	
	*n (*%*)*	*n (*%*)*	*n (*%*)*	
Sex (female)	87 (94.6%)	32 (97.0%)	55 (93.2%)	χ^2^ = 0.58, *p* = 0.65
Missing	0 (0.0%)	0 (0.0%)	0 (0.0%)	*v* = 0.08
Ethnicity				
White British	48 (52.2%)	18 (54.5%)	30 (50.8%)	
Asian British	6 (6.5%)	2 (6.1%)	4 (6.8%)	
Black British	5 (5.4%)	3 (9.1%)	2 (3.4%)	LR = 4.18, *p* = 0.63
White	15 (16.3%)	7 (21.2%)	8 (13.6%)	*v* = 0.21
Mixed	5 (5.4%)	1 (3.0%)	4 (6.8%)	
Other	2 (2.2%)	0 (0.0%)	2 (3.4%)	
Missing	11 (12.0%)	2 (6.1%)	9 (15.3%)	
ICD-10 Diagnosis				
AN	29 (31.5%)	10 (30.3%)	19 (32.2%)	
AN-R	48 (52.2%)	18 (54.5%)	30 (50.8%)	LR = 0.12, *p* = 0.94
AN-BP	15 (16.3%)	5 (15.2%)	10 (16.9%)	*v* = 0.04
Missing	0 (0.0%)	0 (0.0%)	0 (0.0%)	
	* **Mean (SD)** *	* **Mean (SD)** *	* **Mean (SD)** *	
Weight				
Kilograms	42.28 (6.59)	41.66 (7.22)	42.62 (6.25)	*z* = 0.53, *p* = 0.59, *r* = 0.06
%mBMI	80.85 (6.75)	80.18 (7.14)	81.22 (6.56)	*z* = −0.54, *p* = 0.59, *r* = −0.06
Missing	0 (0.0%)	0 (0.0%)	0 (0.0%)	
EDE-Q global scores	3.6 (1.5)	3.4 (1.5)	3.8 (1.5)	*z* = 0.62, *p* = 0.54, r = −0.06
Missing	8 (8.7%)	1 (3.0%)	7 (11.9%)	
RCADS-YP anxiety score	43.5 (21.3)	46 (22.1)	42.9 (20.9)	*t* = −0.83, *p* = 0.41, *d* = −0.19
Missing	13 (14.1%)	4 (12.1%)	9 (15.3%)	
RCADS-P anxiety score	33.8 (19.5)	38.9 (23.8)	31.2 (16.6)	*t* = −1.7, *p* = 0.13, *d* = −0.40
Missing	7 (7.6%)	4 (12.1%)	3 (5.1%)	

Note. *p* < 0.05. Abbreviations: AN = anorexia nervosa—unspecified; AN-R = anorexia nervosa—restricting type; AN-BP = anorexia nervosa—binge purge type. EDE-Q = Eating Disorder Examination Questionnaire. LR = likelihood ratio test. RCADS-YP = the Revised Child Anxiety and Depression Scale- Young Person Version; RCADS-P = Revised Child Anxiety and Depression Scale—Parent Version. SD = standard deviation.

**Table 2 nutrients-16-04117-t002:** Baseline predictors of dietetic input during FT-AN treatment.

Predictor Variables	B	S.E.	Wald	*df*	*p*	Exp (B)	95% CI Lower	95% CI Upper
EDE-Q scores	−0.16	0.22	0.56	1	0.46	0.85	0.56	1.30
RCADS-YP anxiety scores	0.01	0.02	0.19	1	0.67	1.01	0.98	1.04
RCADS-P anxiety score	0.02	0.01	2.72	1	0.10	1.02	1.00	1.05
Baseline weight (%mBMI)	−0.05	0.04	1.52	1	0.22	0.95	0.88	1.03

Note: Analysis included 70 cases (dietetic input, *n* = 23; no dietetic input, *n* = 47). Abbreviations: EDE-Q = Eating Disorder Examination Questionnaire; RCADS = the Revised Child Anxiety and Depression Scale; RCADS-P = Revised Child Anxiety and Depression Scale—Parent Report; RCADS-YP = Revised Child Anxiety and Depression Scale—Young Person Report.

**Table 3 nutrients-16-04117-t003:** Comparison of weight outcomes of groups based on receiving dietetic input during FT-AN treatment and not receiving dietetic input during FT-AN treatment.

		Dietetic Input (*n* = 33)		No Dietetic Input (*n* = 59)		Test Statistic for %mBMI
		Kilograms	%mBMI	Kilograms	%mBMI	
Baseline	Mean (*SD*)	41.7 (7.2)	80.2 (7.1)	42.6 (6.3)	81.2 (6.6)	*z* = −0.54, *p* = 0.59 *r* = −0.06
	Missing, *n* (%)	0 (0.0%)		0 (0.0%)		
Early treatment (4–6 week)	Mean (*SD*)	43.0 (7.3)	83.3 (7.7)	45.7 (7.3)	87.3 (7.3)	*z* = −2.14, *p* = 0.03 *r* = −0.22
	Mean early change (*SD*)	+2.3 (3.0)	+4.4 (4.3)	+3.1 (2.2)	+6.3 (4.4)	*z* = −1.81, *p* = 0.07 *r* = −0.19
	*Missing, n (%)*	4 (12.1%)		17 (28.8%)		
Discharge	Mean (*SD*)	49.2 (7.4)	92.3 (7.5)	50.3 (6.0)	93.1 (7.4)	*z* = −0.12, *p* = 0.68 *r* = −0.01
	Mean overall change (*SD*)	+7.5 (5.4)	+12.1 (10.1)	+7.6 (4.4)	+7.6 (4.4)	*z* = −0.01, *p* = 0.94 *r* = −0.001
	*Missing, n (%)*	0 (0.0%)		0 (0.0%)		

*p* < 0.05. Abbreviation: SD = standard deviation. ‘Early change’ captures difference in weight from baseline to early treatment point. ‘Overall change’ captures difference in weight from baseline to point of discharge.

## Data Availability

The original contributions presented in this study are included in the article. Further inquiries can be directed to the corresponding author.

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
