# Peer review of "An Exploration of Clinical Characteristics and Treatment Outcomes Associated with Dietetic Intervention in Adolescent Anorexia Nervosa"

_nutrients, 2024, doi:10.3390/nu16234117_

Round 1

Reviewer 1 Report

Comments and Suggestions for Authors

Abstract

1. While the study’s focus on dietitians' roles in Family Therapy for anorexia nervosa (FT-AN) is clear, a slight refinement of the purpose statement could further differentiate between identifying characteristics of adolescents needing dietetic intervention and assessing treatment outcomes. 

2. The introduction briefly touches on the debate surrounding dietitians' roles in FT-AN, but it needs to establish this gap's significance adequately in existing research. Doing so will strengthen the rationale.

3. The retrospective cohort design is appropriate given the study's objectives, which are essential in the field. However, the sample size, a critical factor in assessing the study's statistical power and generalizability, must be mentioned. This information is vital to fully understanding the study's potential impact and should be included.

4. Using logistic regression without finding significant predictors raises questions about whether it was the most suitable analysis for this sample. 

5. Reporting effect sizes alongside p-values would enhance interpretation by indicating the clinical relevance of the findings.

6. The abstract is well-structured chiefly, but some sentences can be condensed for clarity, especially in the methodology and results sections.

Introduction

1. The introduction is well-organized, starting with an overview of AN and then progressing to the role of FT-AN and dietitians within this context. However, it can benefit from a more focused statement that clearly defines the purpose and unique contribution of the current study.

2. The background on AN and the importance of early weight gain is informative but can be streamlined. For example, the introductory section can focus more on the necessity of dietetic interventions rather than reiterating general information on weight restoration in AN.

3. Including more background on FT-AN and FBT earlier on can help readers unfamiliar with these terms understand their differences and the relevance of dietetic interventions.

4. The text can benefit from a more critical analysis of studies discussing limitations or inconsistent findings of the studies and guidelines cited. This would emphasize the need for further research in clinical framework and standard practices.

5. There is a need for a more straightforward summary of evidence gaps in dietetics' role in adolescent FT-AN, mainly as previous research primarily focuses on adult populations.

6. The terms “Family Therapy for Anorexia Nervosa” (FT-AN) and “Family-Based Therapy” (FBT) are well-defined, but introducing them in a more compact form might enhance readability.

7. Some language can be more concise. For example, phrases like "Dietitians are skilled in the assessment and management of malnutrition, disordered eating patterns, and nutrition-related deficiencies" can be condensed to focus on dietitians' specific contributions to FT-AN.

8. The section comparing FT-AN and FBT is informative but lengthy. I suggest adding a table or a more concise comparison to help the reader understand the different roles of dietitians in each approach.

9. The section mentioning gaps in evidence for dietetic interventions in adolescent AN can be more assertive in highlighting the importance of understanding dietitians' roles in FT-AN, as opposed to other therapy models.

10. Expanding on why adolescent AN treatment differs from adult treatment, especially regarding family dynamics, would strengthen the study’s justification.

11. The shift from discussing dietetics' general role in AN treatment to the need for research in FT-AN needs to be smoother. Explicitly connecting the need for evidence-based guidance on dietetic interventions with the study's aim would improve readability.

12. The authors briefly mention that existing studies show variability in dietetic outcomes. Still, providing examples or specific studies that highlight this inconsistency would be beneficial.

13. The discussion about the MDT role is scattered. Emphasizing the collaborative nature of the MDT earlier in the introduction would provide a more precise context for dietitians' contributions.

14. The study’s significance is implicit but can be made more explicit by emphasizing how a better understanding of dietetic interventions in adolescent FT-AN can influence treatment protocols or guidelines.

Methods

1. The ethical approval is appropriately documented; however, it would strengthen the section to briefly mention the ethical considerations involved, particularly given the study's retrospective design and the vulnerable adolescent population.

2. While the authors use a retrospective cohort design, further justification is needed to choose this design. It would help readers understand why this approach was suitable for investigating the role of dietetic input in FT-AN treatment.

3. The exclusion criteria, especially "not clinically underweight (>95% median BMI)" and "dropped out of treatment," are confusing. Define terms such as "tier 4 ED services" and clarify if this excludes those with AN in different severities.

4. It is essential to mention that dietetic referrals were based on clinical judgment rather than standardized criteria, but this introduces potential selection bias. The authors should discuss/ explain how this may impact the generalizability and interpretation of results.

5. Since the data collector was not blinded to the study aims, there is potential for bias in data extraction. Discuss whether measures were taken to minimize this bias (e.g., inter-rater reliability checks or cross-validation).

6. No details are provided about how often weights were recorded or any potential variances in measurement protocol (e.g., time of day, clothing, etc.). This might introduce inconsistency, affecting the accuracy of weight change measurements.

7. It is unclear if baseline assessments were administered uniformly or varied at the start of treatment. Specify this to improve replicability.

8. Multiple statistical tests (e.g., Mann-Whitney U, independent t-tests, ANOVA) with relatively small subgroups (dietetic vs. non-dietetic) increase the chance of type I error. The authors should mention whether any corrections for multiple testing were applied.

9. The rationale for assessing patients at baseline, 4-6 weeks, and discharge needs to be explained. It would be helpful to clarify why these specific time points were chosen.

10. The flowchart does not clearly distinguish between reasons for exclusion. Using consistent phrasing and simplifying the structure may improve readability.

11. The authors ‘reasons for dietetic referrals, like “increasing oral intake” and “weight maintenance,” are too general. They should provide more detail on these categories or illustrate them with examples.

Results

1. The lack of significance in treatment duration might imply that adding dietetic input does not alter overall treatment length. However, it is still being determined if there is a clinical rationale for more sessions in the dietetic group. Adding qualitative or observational data regarding the content and perceived effectiveness of the additional sessions could provide more depth.

2. The significant difference in early treatment weight outcomes between groups (dietetic group showing lower %mBMI) is noted, but there is no discussion regarding the potential reasons for this difference. This could imply pre-existing differences between groups or differing treatment responses that merit further analysis. The authors should consider including covariates or additional subgroup analysis that might help determine if pre-existing characteristics influenced the discrepancy in early treatment weight gain.

3. The lack of a significant interaction effect between weight and group suggests that the added dietetic input does not affect weight gain across treatment points. 

4. The non-significant association between timing and early weight gain is not interpreted within the clinical context. It needs to be clarified if timing differences in dietetic input have any theoretical basis for influencing early treatment outcomes.

5. Effect sizes (e.g., d-values) are only reported selectively.

Discussion

1. The lack of predictive factors is noted, but the discussion could delve deeper into possible explanations for this result, particularly considering the role of clinical judgment in determining dietetic input. The authors can hypothesise why these baseline factors do not predict dietetic input (e.g., variability in clinician decision-making or differences in initial patient assessments). This would add depth to this section.

2. While the discussion suggests that additional dietetic input helped align weight outcomes by discharge, it needs an analysis of why dietetic support is associated with slower initial progress. This may raise questions about the immediate impact of dietetic input in early treatment stages.

3. Although the therapeutic alliance is mentioned, the role of specific dietetic interventions in strengthening this relationship is not thoroughly explored.

4. The authors do not address the nuances of decision-making across different stages of treatment despite acknowledging that early weight gain alone does not drive dietetic referrals.

5. The benefits of direct dietetic intervention are noted, but what about indirect interventions (such as meal plans)? 

6. The comparison with adult ED studies can benefit from a clearer explanation of developmental differences between adolescents and adults with AN, which might influence dietetic intervention outcomes.

7. The authors can expand on how missing data or reliance on weight data can skew interpretations, especially for patients who did not have a final weight recorded.

8. The authors mention the need for patient and caregiver perspectives but do not address how these perspectives might offer unique insights into dietetic support's effectiveness.

9. The concluding statements about equivalent weight outcomes may underplay the value of early dietetic input in achieving these results, given the increased treatment intensity required. Reframing the conclusion to recognize that dietetic input may support patients struggling with early weight gain while highlighting its added value in maintaining equivalent final outcomes would make a stronger case for its inclusion in treatment.

Comments on the Quality of English Language

I suggest an English editing 

Reviewer 2 Report

Comments and Suggestions for Authors

This study on the role of dietitians in Family Therapy for Anorexia Nervosa (FT-AN) offers insightful but limited findings. 

The study does not clarify the specifics of the dietetic interventions provided to participants, such as the content or structure of sessions, or whether these interventions followed a standardized protocol. This gap makes it difficult to assess the consistency and comparability of dietetic support across cases. Without a clear framework for dietetic intervention, drawing meaningful conclusions about its efficacy becomes challenging.

The retrospective cohort design and unspecified sample size limit the generalizability of the findings. Retrospective studies are vulnerable to biases, such as selection bias, and may not capture the full complexity of patient needs or outcomes. Additionally, the study lacks information about potential confounding variables that could impact weight restoration outcomes, such as prior treatment history, comorbid mental health conditions, or other support services accessed outside of FT-AN.

Although weight restoration is a critical outcome in FT-AN, the study does not consider other important aspects of recovery, such as improvements in cognitive, emotional, and behavioral symptoms of anorexia. Focusing solely on weight outcomes overlooks dietitians’ potential contributions to cognitive or emotional recovery, which can be equally significant in anorexia treatment.

The study found no specific predictors for dietetic input, suggesting a lack of standardized criteria for when dietetic support should be introduced in FT-AN. This ambiguity underscores the need for clearer guidelines or assessment tools to help clinicians identify patients who may benefit from dietetic involvement early in treatment. Moreover, this inconsistency in the application of dietetic support may explain why baseline characteristics between groups did not differ significantly.

The study’s narrow focus on quantitative outcomes means it misses qualitative insights from patients and their families, who might offer valuable feedback on dietetic support's perceived benefits (or limitations) in FT-AN. Including these perspectives could help clarify how dietetic interventions align with patients' broader recovery goals and support the family-centered nature of FT-AN.

Future studies would benefit from defining specific dietetic interventions and ensuring they are uniformly applied. This approach would improve comparability and help determine whether particular components of dietetic support have unique benefits in FT-AN.

It is essential to examine cognitive, emotional, and behavioral outcomes alongside weight restoration to understand dietitians’ full role in supporting FT-AN. These expanded outcome measures could provide insights into whether dietetic support influences other critical aspects of recovery.

Prospective studies with larger sample sizes would help clarify the temporal relationship between dietetic input and patient outcomes and identify potential predictors for dietetic intervention needs.

Round 2

Reviewer 2 Report

Comments and Suggestions for Authors

After the modifications made in the manuscript, I believe it can be published.